# Systemic Immune and Tumor Marker Profiles in Ovarian and Deep Infiltrating Endometriosis: Associations with Disease Severity and Symptom Burden

**DOI:** 10.3390/ijms26199581

**Published:** 2025-10-01

**Authors:** Tamara N. Ramírez-Pavez, Pilar Marín-Sánchez, Ana Nebot, Laura García-Izquierdo, Lucía Nieto-Meca, Rocío Sánchez, Francisco Machado-Linde, María Martínez-Esparza

**Affiliations:** 1Biochemistry and Molecular Biology (B) and Immunology Department, School of Medicine, IMIB and Regional Campus of International Excellence “Campus Mare Nostrum”, Universidad de Murcia, 30100 Murcia, Spain; t.ramirezpavez@um.es (T.N.R.-P.); ml.garciaizquierdo@um.es (L.G.-I.); fmachado@um.es (F.M.-L.); 2Gynecology and Obstetrics Service, Hospital General Universitario Santa Lucía, 30202 Cartagena, Murcia, Spain; mp.marin@carm.es (P.M.-S.); ana_nebot@hotmail.es (A.N.); 3Gynecology and Obstetrics Service, Hospital Clínico Universitario Virgen de la Arrixaca, 30120 Murcia, Spain; lucianietomeca@gmail.com; 4Gynecology and Obstetrics Service, Hospital General Universitario Reina Sofía, 30003 Murcia, Spain; rsgomez@gmail.com

**Keywords:** ovarian endometriosis, deep infiltrating endometriosis, peripheral blood leukocytes, cytokines, tumor markers, inflammation, non-invasive biomarkers, monocytes

## Abstract

Endometriosis is a chronic, estrogen-dependent inflammatory disease with heterogeneous clinical manifestations and uncertain systemic immune involvement. This study aimed to characterize peripheral immune profiles and circulating tumor markers in women with ovarian endometrioma (OE) and deep infiltrating endometriosis (DIE), and to explore their associations with disease severity, symptom burden, and physical health perception. Peripheral blood leukocyte subsets, plasma cytokines, and tumor markers (CA125, CA19-9, CEA, HE4) were analyzed in 146 patients and 50 healthy controls. OE was associated with increased monocyte counts and reduced neutrophil proportions, while DIE showed elevated levels of IL-8 and Galectin-1. IL-33 levels correlated negatively with the revised American Society for Reproductive Medicine (rASRM) scores and positively with neutrophil proportion, suggesting a role in systemic immune regulation. Tumor marker levels varied by subtype: CA19-9 was higher in OE, and CEA in DIE. CA125 correlated with disease severity, and CEA with monocyte levels. Exploratory heatmaps revealed consistent immune-tumor associations linked to anatomical severity and symptom profiles. Although exploratory, these findings highlight the presence of distinct systemic immune patterns in endometriosis and support the potential of integrative blood-based biomarkers for future diagnostic and stratification strategies.

## 1. Introduction

Endometriosis is an estrogen-dependent gynecological disorder characterized by the growth of endometrial-like tissue outside the uterine cavity, most commonly on the peritoneum and ovaries. Clinical manifestations vary depending on lesion location and include chronic pelvic pain, dysmenorrhea, dyspareunia, dyschezia, and dysuria [1]. The disease affects approximately 10% of women of reproductive age, accounting for more than 200 million cases worldwide [2], and represents a major cause of infertility in women under 35 years of age [3]. Beyond its reproductive implications, endometriosis substantially impairs quality of life [4] and generates considerable socioeconomic costs, estimated at around $10,000 per patient annually in high-income countries, largely attributable to work absenteeism [5,6]. Although histologically benign, endometriosis has been linked to an increased risk of certain ovarian cancers [7].

Based on lesion location and depth, endometriosis is classified into three main subtypes: superficial (SE), ovarian (OE), and deep infiltrating endometriosis (DIE) [8]. SE is confined to the peritoneal surface and is usually managed pharmacologically. OE involves ovarian tissue and may appear as cystic or non-cystic lesions, with the cystic form typically referred to as a chocolate cyst. DIE represents the most severe subtype, characterized by lesions infiltrating more than 5 mm beneath the peritoneal surface and/or involving pelvic organs, ligaments, or the rectovaginal septum. These lesions often induce extensive fibrosis, anatomical distortion, and severe symptoms [9]. Although definitive diagnosis requires laparoscopic confirmation [10], clinical evaluation complemented by imaging techniques such as ultrasound or MRI is commonly employed [11].

The pathogenesis of endometriosis is multifactorial, involving genetic, environmental, and immunological factors that support the survival of ectopic endometrial tissue [10]. Increasing evidence highlights endometriosis as an inflammatory disorder with altered monocyte/macrophage function. Macrophages, central regulators of innate and adaptive immunity, accumulate in the peritoneal cavity in response to chemotactic signals [12,13]. Under physiological conditions, they efficiently clear endometrial debris; however, in endometriosis this clearance appears compromised [14]. Immune dysregulation is further reflected by increased numbers of activated macrophages, T and B lymphocytes, and elevated levels of pro-inflammatory cytokines [15], alongside altered Th1/Th2 ratios [16], reduced natural killer (NK) cell cytotoxicity [17] and upregulated Killer Immunoglobulin-like Receptor (KIR) expression in NK cells [18].

Evaluation of disease severity typically follows two complementary approaches. Anatomical severity is most commonly assessed using the revised American Society for Reproductive Medicine (rASRM) classification, which assigns a numerical score based on lesion size, depth, location, and adhesions. This scoring system enables both severity grading and comparability across studies, categorizing endometriosis into four stages: Stage I (minimal, 1–5 points), Stage II (mild, 6–15 points), Stage III (moderate, 16–40 points), and Stage IV (severe, >40 points). Although the Enzian classification provides greater anatomical precision, particularly for DIE, its complexity has limited its widespread use in research [19,20].

In parallel, symptom burden is increasingly recognized as a critical dimension of disease severity, since anatomical stage does not always correlate with clinical manifestations. Many women with minimal or mild rASRM staging report debilitating symptoms, whereas others with extensive lesions may remain minimally symptomatic [21]. Pain intensity and quality of life indicators are therefore essential to capture the true clinical impact of the disease. Notably, physical health perception and functional impairment represent key complementary markers of disease burden. Recent evidence, such as the study by Marín Sánchez et al. (2025) [22], shows that women with DIE report significantly lower physical health-related quality of life than those with isolated OE, despite similar levels of reported pain.

To address this clinical dissociation, the present study adopts a dual framework for evaluating disease severity. Anatomical severity is quantified using the rASRM score, while symptom burden is assessed through two complementary metrics: pain symptoms, evaluated by type and intensity using visual analog scale (VAS) scores, and perceived physical health, measured via the Physical Component Summary (PCS) score of the SF-12v2 Health Survey—a validated instrument for assessing health-related quality of life [23].

Building on this framework, the aim of the present study is to characterize the systemic immune profile of women with endometriosis by measuring peripheral blood leukocyte populations and circulating inflammatory mediators. We further investigate their associations with four clinically relevant variables: disease subtype (OE vs. DIE), anatomical severity (rASRM score), pain symptoms (VAS), and physical health-related quality of life (SF-12v2 PCS score). Additionally, we explore the relationship between immune markers and serum tumor-associated biomarkers (CA125, CA19-9, CEA, HE4) recorded in clinical practice. These exploratory analyses seek to identify immunological patterns with potential value for understanding disease heterogeneity and for guiding the development of non-invasive, biomarker-based diagnostic and stratification tools.

## 2. Results

### 2.1. Clinical Description of Patients: Most Endometriosis Patients Were Stage IV and Presented Deep Infiltrating Lesions

A total of 196 women were included in the study, comprising 50 healthy controls and 146 patients with endometriosis. A summary of clinical and demographic characteristics is presented in Table 1. Detailed values for rASRM score, pain intensity (VAS), physical health status (PCS), and tumor markers are reported and analyzed in subsequent sections.

The mean age was 35 ± 5 years in the control group and 40 ± 8 years in the endometriosis group. All participants presented a body mass index (BMI) within the normal range (24.39 ± 3.27 and 23.74 ± 2.90 kg/m^2^, respectively). Hormonal treatment was reported in 40.00% of healthy women and 69.46% of patients with endometriosis (Table 1).

Endometriosis patients were recruited among those undergoing surgery, which ensured histopathological confirmation of diagnosis and also allowed collection of peritoneal samples for parallel research projects. Within this group, disease severity according to the rASRM classification was mainly stage IV (67.12%), while stages I–III were less represented (3.42%, 6.16%, and 10.27%, respectively). With respect to lesion type, OE were observed in 37.67% of patients, and DIE in 56.85% (Table 1).

Overall, the endometriosis cohort was characterized by a predominance of advanced-stage disease (rASRM stage IV) and DIE, representing a clinically complex population suitable for investigating systemic immune alterations.

### 2.2. Associations Between Immune Parameters and Clinical Variables

After describing the clinical and demographic characteristics of the cohort, we next examined the relationships between systemic immune parameters and key clinical variables to explore potential immunological patterns associated with disease severity and symptom burden.

#### 2.2.1. Peripheral Leukocyte Subsets and Disease Features: Monocyte and Neutrophil Alterations Are Associated with Endometriosis Subtype and Severity

We first analyzed the associations between peripheral blood leukocyte subsets and four clinical variables: disease subtype, rASRM score, pain symptoms, and perceived physical health.

Peripheral blood cell populations were evaluated in patients with endometriosis and healthy controls. Although all values remained within the normal reference range, patients with OE exhibited a significant decrease in PMN proportion (52.6 ± 8.7% vs. 57.7 ± 9.0%, *p* = 0.0357), along with a significant increase in monocyte count (0.60 ± 0.22 vs. 0.49 ± 0.15 × 10^3^/µL, *p* = 0.0172) and monocyte proportion (9.3 ± 2.0% vs. 7.3 ± 1.8%, *p* = 0.0314) compared to healthy women (Figure 1). While similar trends were observed in the DIE group, the differences did not reach statistical significance, suggesting that systemic immune alterations may be more pronounced in OE (Figure 1).

Since absolute PMN counts remain unchanged, the reduced percentage in blood likely reflects a relative shift driven by the increase in monocytes rather than an actual neutrophil depletion.

We next investigated whether blood leukocyte composition was associated with disease severity, quantified using the rASRM score (Figure 2). After adjustment for multiple testing using the Benjamini–Hochberg procedure, the positive correlation between rASRM score and the proportion of circulating PMNs remained statistically significant (Figure 2C). Other leukocyte parameters, including monocyte counts and proportions, showed no statistically significant associations after correction, although a modest trend toward a negative correlation was observed for monocyte proportion.

We also analyzed perceived physical health, assessed using the PCS score from the SF-12v2 survey; however, no significant associations were found with any leukocyte parameters.

We then explored potential associations between leukocyte profiles and patient-reported pain symptoms, including dysmenorrhea, chronic pelvic pain, dyschezia, dysuria, and dyspareunia (Table 2).

Among the different pain types analyzed, no correlations with leukocyte profiles remained significant after adjustment for multiple testing. A nominal inverse association was observed between the proportion of circulating monocytes and the intensity of dysuria (*r* = −0.2035, *p* = 0.0258) but this did not withstand FDR correction. Similarly, trends toward higher total leukocyte count in patients reporting more intense dysmenorrhea (*r* = 0.1625, *p* = 0.0842) and dyschezia (*r* = 0.1750, *p* = 0.0569) did not reach significance (Table 2).

#### 2.2.2. Circulating Inflammatory Mediators and Clinical Parameters: Specific Cytokine Signatures Distinguish Endometriosis Subtypes and Correlate with Disease Severity and Pain

We then extended the analysis to soluble inflammatory mediators to further characterize the systemic immune landscape in endometriosis.

Patients with endometriosis exhibited elevated plasma levels of the pro-inflammatory chemokine IL-8 and the anti-inflammatory lectin GAL-1, along with decreased levels of sCD163, a monocyte/macrophage activation marker. When stratified by disease subtype, IL-8 levels (Figure 3A) remained significantly increased in the DIE group (13.4 [9.6–20.4] pg/mL vs. 7.4 [5.2–15.7] pg/mL, *p* = 0.0143). The OE group showed a comparable IL-8 distribution (11.8 [7.0–24.7] pg/mL), but the difference relative to controls did not reach statistical significance (*p* = 0.2619).

For GAL-1 (Figure 3B), increased levels were observed exclusively in the DIE group (8.7 [6.1–14.8] ng/mL vs. 6.5 [3.8–8.0] ng/mL, *p* = 0.0216), whereas values in the OE group (6.5 [3.8–9.6] ng/mL) were comparable to controls (*p* > 0.9999). In contrast, the reduction in sCD163 levels (Figure 3C) was most pronounced in the OE group (332.1 [244.9–383.0] ng/mL vs. 573.2 [404.3–792.4] ng/mL, *p* = 0.0025), while the DIE group showed a wider distribution (420.1 [272.4–709.5] ng/mL) without significant differences from controls (*p* = 0.2844).

No significant differences were observed in the remaining cytokines analyzed. Pro-inflammatory markers (TNF-α, IL-1β), anti-inflammatory cytokines (TGF-β, IL-10), and dual-function mediators (IL-33, GAL-3) displayed comparable levels across groups.

We next examined whether circulating cytokine levels were associated with disease severity (rASRM), physical health status (PCS), or pain symptoms.

As shown in Table 3, a nominal inverse correlation was observed between IL-33 with the rASRM score (*r* = −0.4107, *p* = 0.0269), suggesting that lower IL-33 levels may be associated with more extensive anatomical involvement. However, this association did not remain significant after adjustment for multiple testing. Other inflammatory mediators, including TNF-α, IL-1β, IL-8, IL-10, TGF-β, GAL-1, GAL-3, and sCD163, did not correlate significantly with rASRM score (Table 3).

Regarding physical health, TNF-α was inversely correlated with PCS (*r* = −0.4937, *p* = 0.0142) and this association remained significant after adjustment for multiple testing, suggesting that higher systemic inflammation may be associated with poorer physical health perception. No other cytokines showed significant associations (Table 4).

In the analysis of pain symptoms, several cytokines and soluble mediators showed nominal correlations with specific pain domains, including TNF-α with dysmenorrhea, sCD163 with dyschezia, GAL-1 with dyspareunia, IL-33 with dyschezia, and GAL-3 with chronic pelvic pain (Table 5). However, none of these associations remained significant after adjustment for multiple testing. These findings should therefore be interpreted as exploratory and hypothesis-generating rather than conclusive evidence of immune–pain relationships.

### 2.3. Associations Between Tumor Markers and Immune Parameters

Although not considered immune mediators, circulating tumor markers may reflect systemic responses or lesion burden. We therefore assessed their associations with both immune and clinical parameters. These analyses aimed to investigate whether tumor marker levels track with systemic immune alterations.

#### 2.3.1. Tumor Markers and Leukocyte Populations: CEA Levels Correlate with Leukocyte and Monocyte Counts in Endometriosis Patients

We next examined whether circulating levels of tumor-associated markers correlated with peripheral blood leukocyte profiles in patients with endometriosis. As shown in Table 6, total leukocyte counts correlated positively with CEA levels (*r* = 0.2886, *p* = 0.0086), and this association remained significant after FDR adjustment. This association was primarily driven by monocytes, as CEA also correlated significantly with monocyte counts (*r* = 0.2626, *p* = 0.0172) which also persisted after multiple testing correction. In contrast, HE4 showed no significant associations with specific leukocyte subsets (*r* = 0.1947, *p* = 0.0629). No correlations were observed between leukocyte parameters and CA125 or CA19-9 levels (Table 6). These findings suggest that CEA, and to a lesser extent HE4, may reflect subtle immune activation or monocyte involvement in a subset of endometriosis patients.

#### 2.3.2. Tumor Markers and Circulating Inflammatory Mediators

We next examined the association between tumor marker levels and plasma inflammatory mediators. A significant positive correlation was observed between IL-8 and both CA125 (*r* = 0.3011, *p* = 0.0091) and CA19-9 (*r* = 0.3172, *p* = 0.0059). Additionally, IL-33 showed a strong inverse correlation with CA125 (*r* = −0.5559, *p* = 0.0026). All of these associations remained significant after FDR adjustment. No other significant associations were observed among the inflammatory mediators analyzed (Table 7).

### 2.4. Associations Between Tumor Markers and Clinical Variables: CA125, CA19-9 and HE4 Show Selective Associations with Pain and Disease Subtype in Endometriosis

We also explored the clinical relevance of tumor markers by assessing their association with anatomical severity, symptom burden, and disease subtype in endometriosis patients. Findings from these analyses may contribute to a better understanding of the diagnostic and prognostic value of tumor markers in endometriosis.

We first examined whether tumor marker levels varied according to the type of endometriosis. As shown in Figure 4, CA19-9 levels were significantly higher in patients with OE compared to those with DIE (40.8 ± 39.8 U/mL vs. 27.2 ± 34.2 U/mL, *p* = 0.0439; Figure 4B), whereas CEA levels were significantly higher in DIE (1.5 ± 1.6 ng/mL vs. 0.8 ± 0.6 ng/mL, *p* = 0.0104; Figure 4C). No statistically significant differences were observed for CA125 or HE4 levels between subtypes (Figure 4A,D).

We next investigated the relationship between tumor markers and disease severity assess using the rASRM score. A nominal positive correlation between CA125 levels and the rASRM score was found (*r* = 0.2127, *p* = 0.0493); however, this association did not remain significant after FDR adjustment (Figure 5A). CA19-9, CEA, and HE4 showed no significant associations correlations (Figure 5B–D).

We also evaluated whether tumor marker levels were associated with perceived physical health (PCS score). None of the markers showed significant correlations with PCS, although CA125 showed a non-significant positive trend (*r* = 0.2066, *p* = 0.1015, *n* = 64).

Finally, we assessed associations between tumor markers and patient-reported pain symptoms (Table 8). Several nominal correlations were observed, including negative correlation of CA125 with dyschezia (*r* = −0.2113, *p* = 0.0321), CA19-9 with chronic pelvic pain (*r* = −0.2161, *p* = 0.0408) and dyspareunia (*r* = −0.2387, *p* = 0.0251), and a positive correlation between HE4 and dysmenorrhea intensity (*r* = 0.2211, *p* = 0.0420). However, only the negative correlation of CA19-9 with dyspareunia remained significant after FDR adjustment. No significant associations were observed for CEA with any of the clinical variables analyzed.

### 2.5. Summary of Relevant Correlations

To provide an integrated and visual overview of the associations identified in this study, we generated a heatmap summarizing correlations between immune parameters and clinical variables. This heatmap incorporates all statistically significant (*p* < 0.05) and trend-level (*p* < 0.1) correlations, while also indicating which associations remained significant after multiple testing correction (FDR < 0.05) (Figure 6).

Among leukocyte parameters, the proportion of circulating neutrophils (PMN %) showed a positive correlation with disease severity, as assessed by the rASRM score (*r* = 0.2265, *p* = 0.0056). This association remained significant after FDR correction. In contrast, the nominal inverse correlation between monocyte proportion and dysuria intensity (*r* = −0.2035, *p* = 0.0258), as well as the trend with chronic pelvic pain (*r* = −0.1600, *p* = 0.0863), did not withstand correction and should be interpreted as exploratory.

Regarding cytokines, TNF-α levels were inversely correlated with PCS scores (*r* = −0.49, *p* = 0.0142), and this association also survived FDR correction, suggesting that higher systemic inflammation is robustly linked to poorer physical health perception. TNF-α additionally showed a nominal positive correlation with dysmenorrhea (*r* = 0.2065, *p* = 0.0403), consistent with a potential role in pain sensitization, although this did not remain significant after correction.

Other cytokines displayed exploratory associations with specific pain domains. For example, IL-33 showed nominal negative correlation with rASRM score (*r* = −0.4107, *p* = 0.0269) and positive association with dyschezia (*r* = 0.4172, *p* = 0.0340), while GAL-1 correlated with dyspareunia (*r* = 0.3963, *p* = 0.0407), sCD163 with dyschezia (*r* = 0.3336, *p* = 0.0330), and GAL-3 negatively with chronic pelvic pain (*r* = −0.3762, *p* = 0.0405). None of these associations survived FDR correction, underscoring their exploratory nature.

Together, these findings highlight the complexity and heterogeneity of systemic immune responses in endometriosis. While only two correlations (PMN %–rASRM and TNF-α–PCS) remained robust after correction, several nominal associations suggest potential links between immune parameters, disease severity, and pain phenotypes that warrant validation in larger cohorts.

To explore whether circulating tumor markers reflect clinical severity or systemic immune alterations in endometriosis, we generated a second heatmap summarizing statistically significant and trend-level correlations between tumor markers and clinical or immune variables (Figure 7).

Among the tumor markers analyzed, CEA showed the most consistent immunological associations. It correlated positively with both total leukocyte count (*r* = 0.2886, *p* = 0.0086) and monocyte count (*r* = 0.2626, *p* = 0.0172), and these associations remained significant after FDR correction, suggesting a robust link between systemic myeloid activation and CEA levels in a subset of patients.

Regarding inflammatory mediators, IL-8 was positively correlated with CA125 (*r* = 0.32, *p* = 0.0049) and CA19-9 (*r* = 0.39, *p* = 0.0011), while IL-33 showed a negative correlation with CA125 (*r* = −0.56, *p* = 0.0026). All three associations remained significant after FDR correction, further reinforcing their biological plausibility.

In contrast, other tumor marker–clinical associations were only nominal. For example, CA125 showed a weak positive correlation with rASRM score (*r* = 0.2127, *p* = 0.0493) and a negative association with dyschezia (*r* = −0.2113, *p* = 0.0321); CA19-9 correlated negatively with chronic pelvic pain (*r* = −0.2161, *p* = 0.0408) and dyspareunia (*r* = −0.2387, *p* = 0.0251), with the latter remaining statistically significant after FDR correction; and HE4 was nominally associated with dysmenorrhea (*r* = 0.2211, *p* = 0.0420). All other associations did not survive FDR correction and should therefore be interpreted as exploratory.

Taken together, these findings indicate that specific tumor markers, particularly CEA and the relationships between IL-8/IL-33 and CA125/CA19-9, may reflect systemic immune activation in endometriosis. Nevertheless, their individual diagnostic value remains limited, underscoring the need for multi-parametric biomarker approaches.

## 3. Discussion

Our study demonstrates that systemic immune alterations in endometriosis are reflected in peripheral blood leukocyte composition, circulating inflammatory mediators, and tumor marker profiles, with distinctive patterns depending on disease subtype. These findings reinforce the concept that endometriosis is not a uniform entity but rather encompasses immunologically diverse phenotypes with potential implications for pathophysiology, clinical manifestations, and biomarker development [1].

In our cohort, most patients presented advanced-stage disease (rASRM stage IV) or DIE. This distribution reflects the clinical profile of women undergoing laparoscopic surgery in our centers, where surgical intervention is generally indicated for severe disease, and also responds to the design of parallel studies in which peritoneal samples and lesions were analyzed. Although this predominance may limit the generalizability of our findings to earlier disease stages, it provides a clinically homogeneous and well-characterized population, particularly suitable for systemic immune and biomarker exploration. Importantly, the majority of these patients were under hormonal treatment, mirroring real-world clinical management. Assessing potential circulating biomarkers in advanced disease may therefore offer valuable insights into their robustness under treatment and their potential applicability for disease monitoring across different stages in future studies.

We observed an increase in circulating monocytes and a relative reduction in neutrophil proportion in patients with endometriosis, particularly in those with OE. Since absolute neutrophil counts remained unchanged, this reduction likely reflects a relative shift driven by monocyte expansion rather than neutrophil depletion. Such alterations are consistent with the systemic footprint of chronic peritoneal inflammation. Previous studies have reported increased circulating non-classical monocytes in endometriosis patients [24] and elevated neutrophil counts typically linked to chronic inflammatory states [25], but our data suggest that the dominant systemic feature may be monocyte expansion rather than neutrophilia. The more pronounced alterations in ovarian endometriomas highlight subtype-specific immune signatures, supporting the notion of disease heterogeneity [1].

When examining soluble mediators, we identified a distinct profile for each subtype. Patients with DIE exhibited increased IL-8 and GAL-1 levels, suggesting an inflammatory milieu that is partially counterbalanced by immunoregulatory signals. In contrast, OE patients showed reduced plasma sCD163 despite higher circulating monocytes, suggesting impaired monocyte/macrophage regulatory activity. sCD163 is a well-recognized marker of macrophage activation and an anti-inflammatory scavenger receptor [26], and its decrease in this context may reflect dysfunctional regulation of innate immune responses. These findings underscore that different lesion types are associated with divergent systemic immune signatures: a mixed inflammatory-regulatory environment in deep lesions versus a more pro-inflammatory, deregulated profile in ovarian endometriosis.

The correlation analyses revealed that only a limited set of associations remained significant after FDR correction, notably the positive association between neutrophil proportion and rASRM score, the inverse correlation between TNF-α and PCS, and the links between CEA and leukocyte/monocyte counts. Additional robust associations were found between IL-8 and CA125/CA19-9, and between IL-33 and CA125. These findings provide consistent evidence that systemic myeloid activation and selected cytokine–tumor marker interactions are reproducible features of endometriosis. Other correlations with pain domains (e.g., monocyte proportion with dysuria, IL-33 with dyschezia, CA19-9 with pelvic pain, HE4 with dysmenorrhea) did not withstand multiple testing correction and should be interpreted as exploratory signals requiring validation.

With regard to disease severity and symptoms, we found that neutrophil proportion correlated positively with rASRM score, while monocyte parameters showed selective associations with specific pain manifestations, such as an inverse relationship with dysuria intensity. These findings suggest that systemic leukocyte shifts may partly mirror disease progression, although their association with pain symptoms was limited. This supports the view that endometriosis-associated pain is multifactorial and not solely explained by systemic immune parameters, reinforcing the need for integrative approaches [27,28].

IL-33 emerged as mediator of particular interest through its strong inverse correlation with CA125. IL-33 is a context-dependent alarmin with both pro- and anti-inflammatory roles [29,30,31,32]. While previous studies proposed roles for IL-33 in lesion persistence and fibrosis [33] our data point toward its integration with tumor marker profiles rather than with clinical severity per se, highlighting a potential avenue for biomarker combinations that bridge immune and tumor-derived signals.

Although CA125 showed nominal correlations with rASRM score and dyschezia, and CA19-9/HE4 with pain, these associations did not survive FDR correction. By contrast, the correlations of CEA with leukocyte and monocyte counts, and of IL-8/IL-33 with CA125/CA19-9, underscore that tumor markers can capture subtle immunological activity beyond their classical role as oncologic biomarkers. This reinforces the concept that their value in endometriosis lies not as standalone markers but as part of integrated biomarker panels.

The potential impact of hormonal treatment must be considered when interpreting our findings. At the time of sample collection, 69.46% of endometriosis patients and 40.00% of healthy controls were receiving hormonal therapy, primarily for symptom management or contraception, respectively. Hormonal treatment is known to modulate immune cell activity, cytokine expression, and tumor marker levels. Therefore, the widespread use of hormonal therapy in both groups may have attenuated certain inflammatory signals and contributed to the relatively modest effect sizes observed. In exploratory stratified analyses, correlations between immune parameters and pain were more evident in untreated patients, whereas they were attenuated in those under treatment, consistent with the therapeutic effects of hormonal therapy. However, the number of untreated patients was small, particularly for cytokine analyses, limiting statistical power. Future studies should include treatment-naïve or stratified cohorts to better disentangle disease-related immune signatures from treatment-induced changes.

To provide a comprehensive overview of the associations identified, we generated two correlation heatmaps. The first summarized the relationships between immune parameters (leukocytes and cytokines) and clinical variables (rASRM score, pain, physical health), confirming distinct immunological patterns associated with severity and symptomatology, with only neutrophil proportion and TNF-α showing robust associations after correction. The second heatmap integrated tumor marker data, revealing consistent correlations between CEA, IL-8/IL-33, and selected immune and tumor parameters, while other tumor marker–symptom associations were only exploratory. These figures reveal potential diagnostic value in combining immune and tumor profiles, especially in distinguishing subtypes and symptom presentations.

Taken together, these findings emphasize the importance of distinguishing robust from exploratory associations in endometriosis biomarker research, and they highlight the need for validation of nominal findings in larger, independent cohorts.

Building on these findings, we propose that future studies explore the development of multivariate models, such as decision tree approaches, integrating blood leukocyte profiles and tumor marker levels to improve screening for endometriosis. Since both monocyte counts and CEA levels can be elevated within normal limits, their combined interpretation may reveal patterns suggestive of endometriosis that would otherwise go unnoticed using standard diagnostic thresholds. HE4, although not robustly associated in our analysis, may still contribute to such models by improving sensitivity in detecting immunologically active disease.

For this strategy to be clinically useful, future studies should include healthy controls and patients with other gynecologic conditions that often mimic endometriosis, such as ovarian cysts. Importantly, treatment status should be considered, as hormonal therapy can reduce tumor marker levels and may obscure immune-related signals. Nevertheless, the persistence of selected immune–tumor marker correlations despite treatment in our cohort supports their potential as subtle indicators of disease activity.

## 4. Materials and Methods

### 4.1. Patients

A prospective study was conducted between October 2014 and March 2025. The control group included 50 women recruited during their family planning consultations, who requested definitive contraception by laparoscopic tubal sterilization. These women had no associated gynecological diseases and were enrolled from the Gynecological Unit of *Hospital General Universitario Santa Lucía* (Cartagena), *Hospital General Universitario Reina Sofía* (Murcia), and the Endometriosis Unit of *Hospital Clínico Universitario Virgen de la Arrixaca* (Murcia), Spain.

The study group consisted in patients with endometriosis aged 18–49, a total of 183 women recruited from the Gynecology Units of *Hospital General Universitario Santa Lucía* in Cartagena, *Hospital General Universitario Reina Sofía* in Murcia, and the Endometriosis Unit of *Hospital Clínico Universitario Virgen de la Arrixaca* in Murcia, Spain. They underwent therapeutic laparoscopy and following surgery, the diagnosis of endometriosis was confirmed by histopathological examination. Based on these results, 146 patients were included in the endometriosis group. The remaining 37 patients were excluded for the following reasons: absence of histopathological confirmation (*n* = 3), diagnosis of endometriosis-associated ovarian carcinoma (*n* = 2), isolated adenomyosis (*n* = 14), uterine fibroids (*n* = 6), or non-endometriotic ovarian cysts (*n* = 12).

General exclusion criteria for the study included the presence of autoimmune or inflammatory diseases, any relevant comorbidities (e.g., chronic metabolic, cardiovascular, or infectious conditions), use of antibiotics other than standard surgical prophylaxis, body mass index higher than 30 kg/m^2^, current oncological conditions, and pregnancy or breastfeeding within the previous six months.

### 4.2. Clinical Data Collection

Clinical and demographic data were obtained from the medical records of all participants using the Selene electronic health record system of the Murcia Health Service (SMS). General variables collected included age, body mass index (BMI), gynecological and obstetric history, menstrual cycle characteristics, date of last menstrual period, preoperative laboratory test results, and postoperative histopathological findings. Hormonal therapy status was documented for all participants, including the type (oral contraceptives, progestins, or levonorgestrel-releasing intrauterine devices) and indication (contraception in controls, symptom management in patients). All patients who were receiving hormonal therapy continued treatment up to the time of surgery, when blood samples were obtained, ensuring that all measurements reflect the ongoing influence of hormonal therapy.

For patients diagnosed with endometriosis, additional clinical variables were retrieved to support disease characterization and stratification. These included: lesion subtype, classified as OE or DIE, based on imaging findings (ultrasound and/or MRI) and surgical reports; disease severity, assessed using the rASRM classification system, which provides a numerical score based on lesion characteristics and adhesion; pain symptoms, recorded from clinical notes, and categorized by type (dysmenorrhea, dyspareunia, dyschezia, dysuria, and chronic pelvic pain) and intensity assessed by asking each patient to rate her pain on a 0–10 visual analog scale (VAS) for each symptom at the time of the clinical visit, following a standardized questionnaire administered by trained staff; and circulating tumor markers routinely measured during follow-up of patients with endometriosis, including cancer antigen 125 (CA125) and 19-9 (CA19-9), carcinoembryonic antigen (CEA), and Human epididymis protein 4 (HE4).

A subset of patients also completed the SF12V2 Health Survey, a validated instrument for assessing health-related quality of life. The survey evaluates eight health domains and yields two summary scores: the Physical Component Summary (PCS) and the Mental Component Summary (MCS) [23]. In this study, only the PCS score was used as an indicator of the perceived physical health impact of the disease; scores range from 0 to 100, with higher values indicating better quality of life.

### 4.3. Human Blood Plasma Collection

For healthy women and endometriosis patients, 3–5 mL of venous blood was collected in K2 EDTA (ethylenediaminetetraacetic acid) vacutainers immediately before surgery. Samples were kept at room temperature and processed within 8 h of collection. Plasma was separated by centrifugation at 1300 rpm (≈300 g) for 3 min at room temperature and the supernatant was carefully transferred into sterile 1.5 mL microtubes in aliquots of 500 µL to avoid repeated freeze–thaw cycles. Aliquots were stored at −80 °C until analysis.

### 4.4. Cytokines Measurements

TNF-α, IL-1β, IL-6, IL-8, IL-10, sCD163, TGF-β1 (Invitrogen, Thermo Fisher Scientific, Carlsbad, CA, USA) and IL-33 (ABclonal, Düsseldorf, Germany), GAL-1 (FineTest, Wuhan, Hubei, China) and GAL-3 (Proteintech, Manchester, England, UK) were quantified in blood plasma by enzyme-linked immunosorbent assay (ELISA) according to the manufacturer’s instructions. The assays were performed in strip plates (Immuno Clear Standard Modules, Thermo Fisher Scientific, Waltham, MA, USA), and the final absorbance at 470 nm and 550 nm was measured by the SPECTROstar Nano (BMG LABTECH, Ortenberg, Germany).

### 4.5. Statistical Analysis

Data obtained in this study were analyzed using GraphPad Prism version 8.3.0 (GraphPad, Boston, MA, USA). Samples with fewer than 30 data points were assumed to have a non-parametric distribution. For variables with more than 30 data points, normality was assessed using the Shapiro–Wilk test (*n* = 30–50), and the Kolmogorov–Smirnov test (n > 50).

For comparisons between two groups, paired or unpaired *t*-tests were used when parametric assumptions were met; otherwise, Mann–Whitney U test were used for unpaired samples.

For comparisons involving more than two groups within a single factor, one-way ANOVA followed by Tukey’s post hoc test was used for parametric paired data. For non-parametric data, the Kruskal–Wallis test followed by Dunn’s post hoc test was applied for unpaired samples.

For correlation analysis Pearson’s or Spearman’s tests were used, depending on the distribution of the data.

Exploratory correlation heatmaps were constructed to visualize relevant associations between systemic immune parameters (circulating leukocyte subsets and plasma inflammatory mediators) and clinical variables, including disease subtype, rASRM score, pain symptoms, PCS score, and tumor marker levels. Only variables with statistically significant (*p* < 0.05) or trend-level (0.05 ≤ *p* < 0.1) correlations were included in the heatmaps.

Heatmaps were generated using Python 3.10, with the pandas, numpy, matplotlib, and seaborn libraries. Values were represented in a diverging color scale centered at zero, and variables were grouped to highlight clinical versus immunological domains.

### 4.6. Use of Language Editing Tools

To improve the clarity and readability of the text, a language model (ChatGPT version GPT-5 mini, OpenAI, San Francisco, CA, USA ) was used for assistance with grammar, phrasing, and editorial refinement.

## 5. Conclusions

In conclusion, our findings support the existence of systemic immune alterations in endometriosis that differ according to lesion subtype and clinical presentation. Although individually modest, these alterations, encompassing leukocyte profiles, cytokines, and tumor markers, form recognizable patterns that reflect disease severity and symptom burden.

The integration of immune and tumor marker data offers a promising direction for developing non-invasive, blood-based tools to aid diagnosis and patient stratification. This is particularly relevant in clinical contexts where imaging results are inconclusive, or laparoscopic confirmation is delayed. By identifying composite immune-tumor signatures, clinicians may eventually distinguish between subtypes or flag patients at higher risk for severe forms of the disease.

While further validation is needed, particularly in treatment-naïve populations and broader gynecologic cohorts, our study provides a proof of concept for the systemic profiling of endometriosis. Given that most patients and a substantial fraction of controls were receiving hormonal therapy, some findings may partly reflect treatment effects rather than disease-specific alterations, highlighting the need for confirmation in untreated cohorts. This approach could support earlier recognition, more tailored management strategies, and better communication with patients regarding the biological basis of their symptoms.

## Figures and Tables

**Figure 1 ijms-26-09581-f001:**
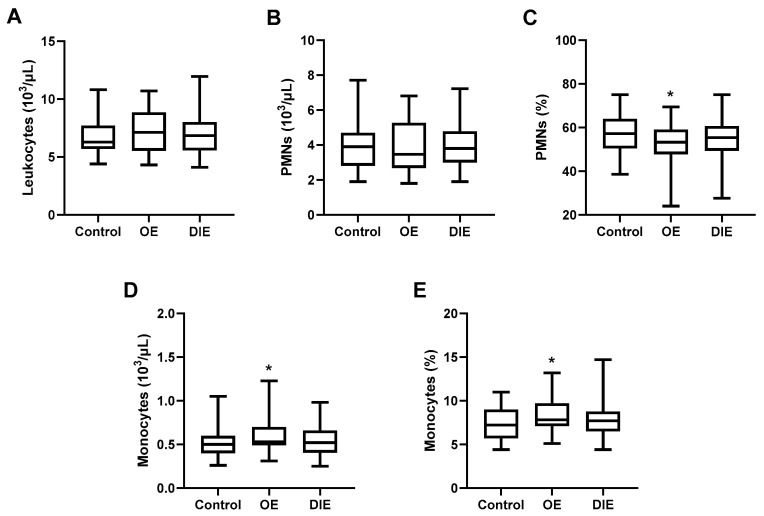
**Differential blood leukocyte profiles across endometriosis subtypes.** Box plots show the distribution of blood leukocyte populations in healthy women (Control, *n* = 47) and patients with stage III–IV endometriosis, stratified into ovarian endometriosis (OE, *n* = 31) and deep infiltrating endometriosis (DIE, *n* = 77). The panels display total leukocyte count (**A**), neutrophil (PMN) count (**B**) and proportion (**C**), and monocyte count (**D**) and proportion (**E**). Minimum and maximum values are indicated. Statistical comparisons were performed using one-way ANOVA with Tukey’s post hoc test. Asterisks denote statistically significant differences versus the control group. * = *p* < 0.05.

**Figure 2 ijms-26-09581-f002:**
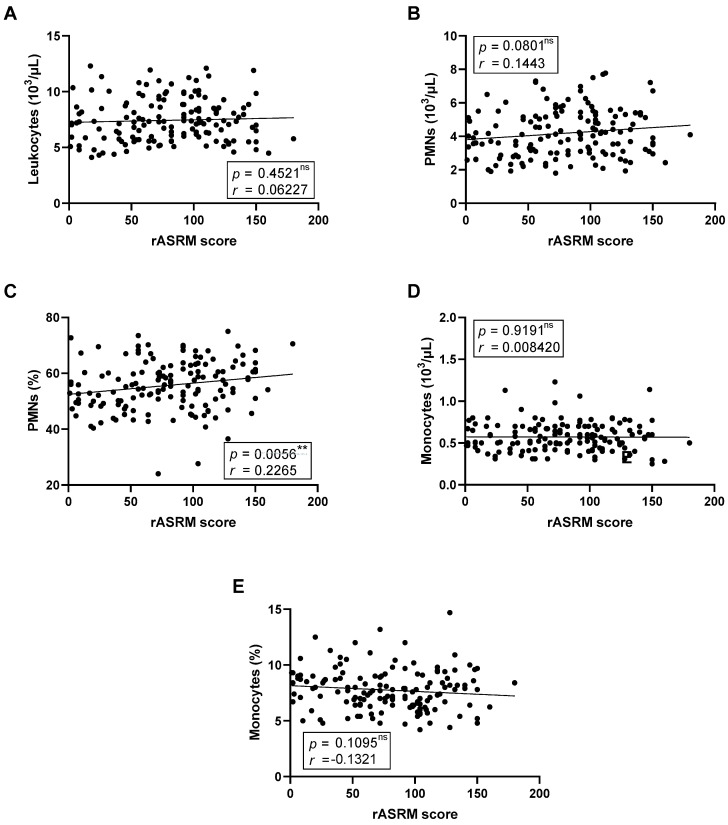
**Correlation between peripheral blood leukocyte distribution and endometriosis clinical severity.** Panels show total leukocyte count (**A**), neutrophil (PMN) count (**B**), PMN proportion (**C**), monocyte count (**D**), and monocyte proportion (**E**) from 146 patients with endometriosis. Correlation coefficients (*r*) and *p*-values are shown. Asterisks indicate nominal significance (unadjusted ** *p* < 0.01). *p*-values surviving multiple testing correction using the Benjamini–Hochberg procedure (FDR < 0.05) are highlighted in bold. ns = non-significant correlations.

**Figure 3 ijms-26-09581-f003:**
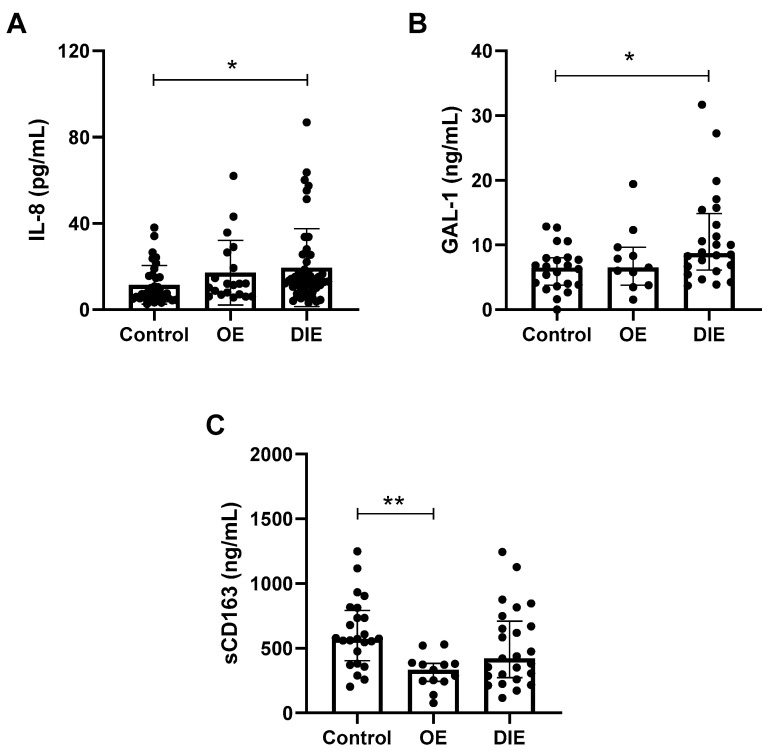
**Differential plasma profiles of inflammatory mediators across endometriosis subtypes.** Graphs show the plasma levels of inflammatory mediators that were altered in patients with endometriosis prior to stratification. The data represent comparisons between healthy women (control group) and patients with stage III–IV endometriosis, further stratified by subtype: ovarian endometriosis (OE) and deep infiltrating endometriosis (DIE). The analyzed mediators include cytokine IL-8 (control: *n* = 35; OE: *n* = 20; DIE: *n* = 49) (**A**), galectin-1 (GAL-1; control: *n* = 22; OE: *n* = 11; DIE: *n* = 24) (**B**), and sCD163 (control: *n* = 24; OE: *n* = 13; DIE: *n* = 25) (**C**). Due to the dispersion and non-Gaussian distribution of the data, results are presented as median and interquartile range (25th–75th percentile). Statistical analyses were performed using Kruskal–Wallis test followed by Dunn’s post hoc test. * = *p* < 0.05; ** = *p* < 0.01.

**Figure 4 ijms-26-09581-f004:**
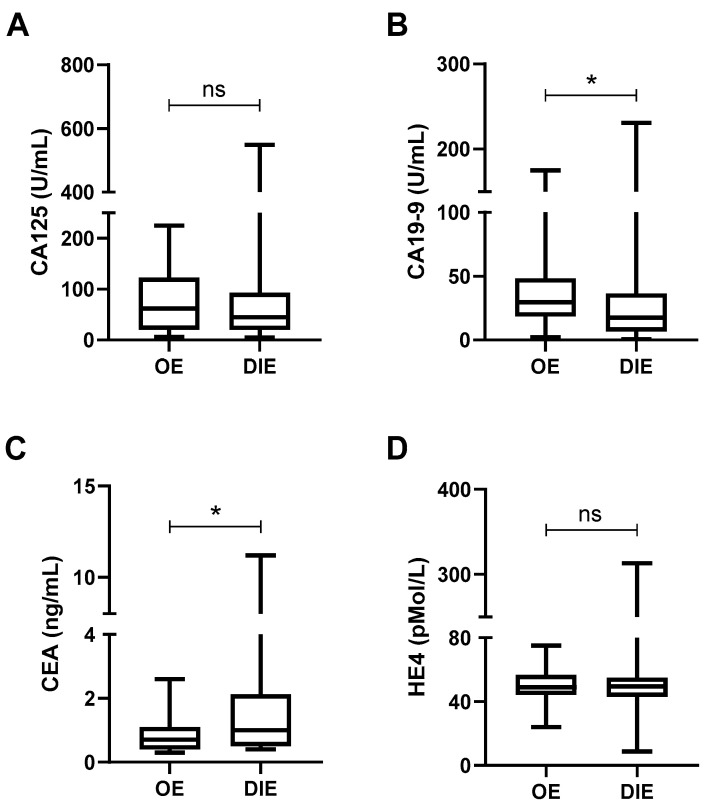
**Differential plasma levels of tumor markers across endometriosis subtypes.** Box plots illustrate the distribution of tumor marker levels in patients with ovarian endometriosis (OE, *n* = 24) and deep infiltrating endometriosis (DIE, *n* = 72). The markers analyzed include CA125 (**A**), CA19-9 (**B**), CEA (**C**), and HE4 (**D**). Statistical comparisons were performed using the *t*-test or Mann–Whitney U test, depending on data distribution. ns = not significant; * = *p* < 0.05.

**Figure 5 ijms-26-09581-f005:**
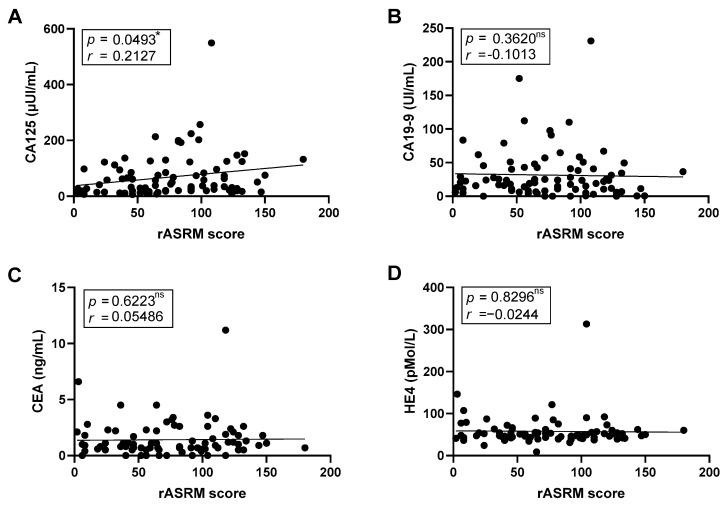
**Correlation between endometriosis severity and tumor marker.** Dot plots illustrating the correlation between disease severity, as assessed by the revised American Society for Reproductive Medicine (rASRM) score, and plasma levels of the tumor markers CA125 (**A**), (*n* = 93), CA19-9 (**B**), CEA (**C**) and HE4 (**D**) (*n* = 91). Correlation coefficients (*r*) and *p*-values were calculated using Pearson’s or Spearman’s tests, according to data distribution. Asterisks indicate nominal significance (unadjusted * *p* < 0.05). ns = not significant correlation.

**Figure 6 ijms-26-09581-f006:**
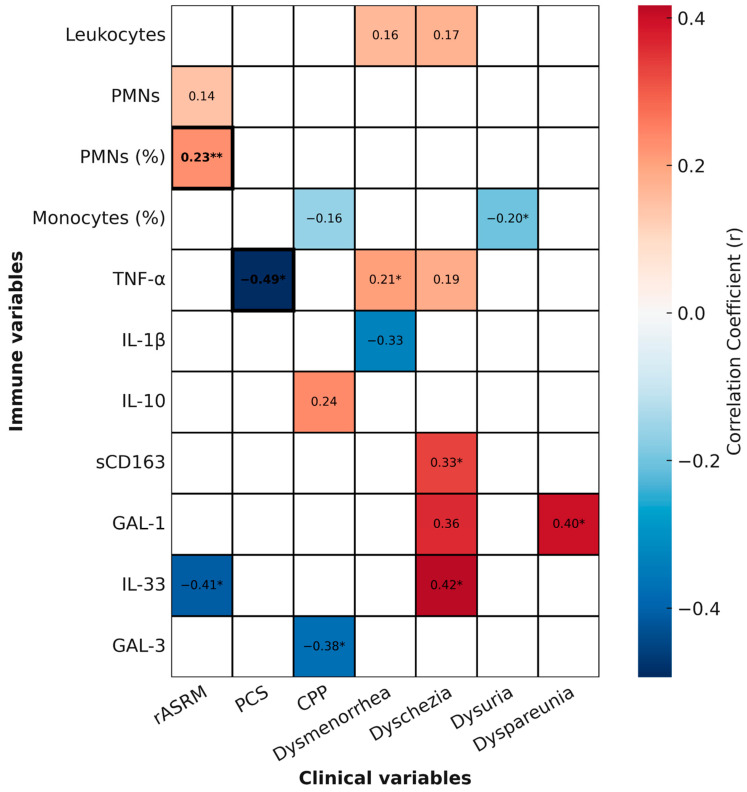
**Correlations between immune parameters and clinical variables in endometriosis.** Heatmap showing relevant correlations between peripheral blood immune parameters-including leukocyte subsets and circulating inflammatory mediators-and clinical measures of endometriosis including anatomical severity assessed with the revised American Society for Reproductive Medicine (rASRM), physical component summary (PCS), and pain symptoms. Red shading indicates positive correlations; blue indicates negative. The intensity of the color reflects the strength of the correlation coefficient (*r*), with darker shades representing stronger associations. Asterisks indicate nominal significance (unadjusted * *p* < 0.05, ** = *p* < 0.01 ), while bold text marks correlations that remained significant after multiple testing correction using the Benjamini–Hochberg procedure (FDR < 0.05). Correlations were calculated using Pearson’s or Spearman’s tests depending on data distribution.

**Figure 7 ijms-26-09581-f007:**
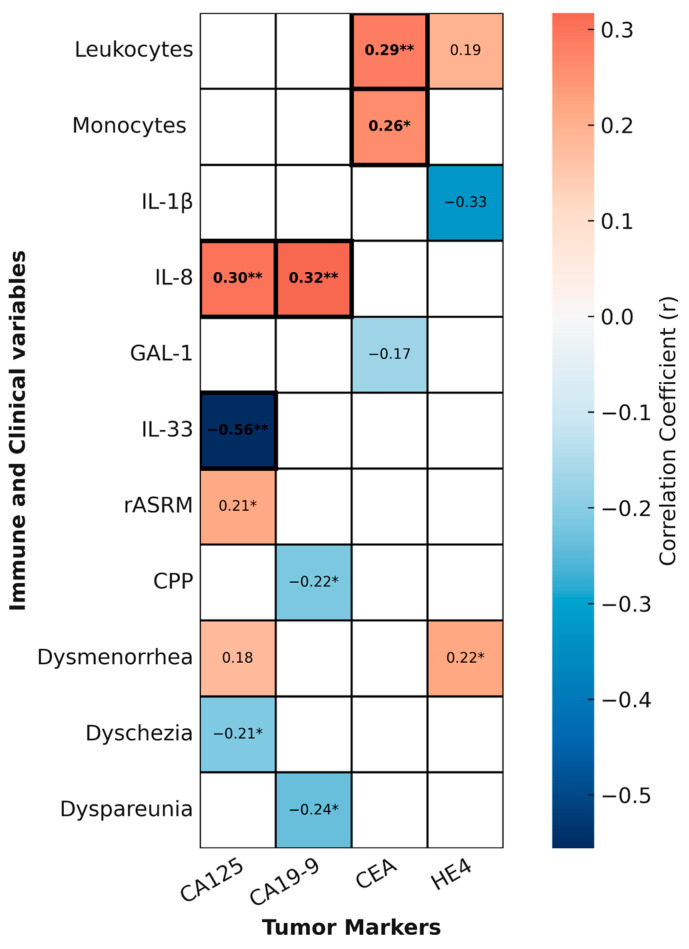
**Correlations heatmap between tumor markers and immune and clinical variables in endometriosis.** Heatmap showing relevant correlation between tumor markers (CA125, CA19-9, CEA, HE4) and selected immune variables (leukocyte counts and circulating inflammatory mediators) and clinical measures as revised American Society for Reproductive Medicine (rASRM) score, Physical Component Summary (PCS) and pain symptoms). Only correlations with *p*  <  0.1 are shown. Red shading indicates positive correlations; blue indicates negative ones. The color intensity reflects the strength of the correlation coefficient (*r*), with darker colors indicating stronger correlations. Asterisks indicate nominal significance (unadjusted * *p* < 0.05, ** *p* < 0.01), while bold text marks correlations that remained significant after multiple testing correction using the Benjamini–Hochberg procedure (FDR < 0.05). Correlations were calculated using Pearson’s or Spearman’s tests depending on data distribution.

**Table 1 ijms-26-09581-t001:** Clinical description of patients included in the study.

Parameter		Healthy Women	Endometriosis
**Number of patients (*n*)**		50	146
**Age (years) ± SD**		35 ± 5	40 ± 8
**BMI (kg/m^2^) ± SD**		24.39 ± 3.27	23.74 ± 2.90
**Hormonal treatment (%)**		40.00	69.46
**Severity** **rASRM ^1^—stage (%)**	I	–	3.42
	II	–	6.16
	III	–	10.27
	IV	–	67.12
**Type of lesions (%)**	OE	–	37.67
	DIE	–	56.85

^1^ The Revised American Society for Reproductive Medicine (rASRM) score classifies endometriosis into four stages based on the severity of the disease. OE = ovarian endometriosis, DIE = deep infiltrating endometriosis.

**Table 2 ijms-26-09581-t002:** Correlation between blood cells distribution and patient-reported pain.

Blood Cells (10^3^/µL)		Dysmenorrhea	CPP	Dyschezia	Dysuria	Dyspareunia
**Leukocytes**	*r*	0.1625	0.1323	0.1750	0.1324	−0.0719
	*p*	0.0842	0.1568	0.0569	0.1494	0.4490
**PMNs**	*r*	0.1307	0.0975	0.1454	0.0414	−0.1498
	*p*	0.1657	0.2977	0.1147	0.6535	0.1113
**Monocytes**	*r*	0.0510	−0.0006	0.0882	0.0106	−0.0831
	*p*	0.5902	0.9946	0.3401	0.9089	0.3814
**Blood cells** **(%)**						
**PMNs**	*r*	0.0453	0.0764	0.0877	−0.0525	−0.1102
	*p*	0.6323	0.4151	0.3428	0.5688	0.2454
**Monocytes**	*r*	−0.1144	−0.1600	−0.0552	−0.2035	−0.0057
	*p*	0.2254	0.0863	0.5509	0.0258 *	0.9523

Spearman’s correlation coefficients (*r*) and corresponding *p*-values (*p*) are shown. Data was obtained from 120 patients with endometriosis. Asterisks indicate nominal significance (unadjusted * *p* < 0.05). CPP = Chronic pelvic pain.

**Table 3 ijms-26-09581-t003:** Correlation between plasmatic inflammatory mediators and rASRM score.

Physiological Effect	Molecule	Patients (*n*)	*r*	*p*-Value
**Pro-inflammatory**	TNF-α	53	0.0371	0.7916
	IL-1β	35	−0.2361	0.1721
	IL-8	74	0.0451	0.7029
**Anti-inflammatory**	TGF-β	35	0.0020	0.9911
	IL-10	57	−0.0261	0.8474
	sCD163	41	0.1905	0.2329
	GAL-1	31	−0.0910	0.6264
**Dual activity**	IL-33	29	−0.4107	0.0269 *
	GAL-3	33	0.0922	0.6099

Correlation coefficients (*r*) and *p*-values were calculated using Pearson’s or Spearman’s tests, according to data distribution. *p*-values (* = *p* < 0.05) were adjusted for multiple comparisons using the Benjamini–Hochberg procedure; significance was determined based on an FDR threshold of 0.05. rASRM = revised American Society for Reproductive Medicine.

**Table 4 ijms-26-09581-t004:** Correlation between plasmatic inflammatory mediators and PCS.

Physiological Effect	Molecule	Patients (*n*)	*r*	*p*-Value
**Pro-inflammatory**	TNF-α	26	−0.4937	**0.0142 ***
	IL-1β	18	−0.0774	0.7600
	IL-8	56	−0.0292	0.8309
**Anti-inflammatory**	TGF-β	23	0.3389	0.1136
	IL-10	41	0.0877	0.5857
	sCD163	25	−0.1508	0.4719
	GAL-1	24	0.0756	0.7253
**Dual activity**	IL-33	21	−0.0740	0.7498
	GAL-3	26	0.3105	0.1227

Correlation coefficients (*r*) and *p*-values were calculated using Pearson’s or Spearman’s tests, according to data distribution. Asterisks indicate nominal significance (unadjusted * *p* < 0.05). *p*-values surviving multiple testing correction using the Benjamini–Hochberg procedure (FDR < 0.05) are highlighted in bold. PCS = Physical component summary.

**Table 5 ijms-26-09581-t005:** Correlation between plasmatic inflammatory mediators and patient-reported pain.

Molecule		Dysmenorrhea	CPP	Dyschezia	Dysuria	Dyspareunia
**TNF-α** **(*n* = 52)**	*r*	0.2065	0.1037	0.1877	0.1446	−0.0136
	*p*	0.0122 *	0.3094	0.0642	0.1532	0.8980
**IL-1β** **(*n* = 44)**	*r*	−0.3305	−0.1265	−0.0882	−0.1977	0.1277
	*p*	0.0603	0.4760	0.6253	0.2702	0.5091
**IL-8** **(*n* = 71)**	*r*	−0.0991	−0.0974	0.0949	0.0044	−0.1072
	*p*	0.4108	0.4125	0.4276	0.9703	0.3805
**TGF-β** **(*n* = 37)**	*r*	−0.1467	0.0701	0.06747	0.1134	0.0761
	*p*	0.3864	0.6846	0.6958	0.5041	0.6589
**IL-10** **(*n* = 50)**	*r*	0.0942	0.2382	−0.0879	0.0579	−0.0639
	*p*	0.5153	0.0923	0.5440	0.6865	0.6697
**sCD163**	*r*	0.1241	0.0116	0.3336	−0.1559	−0.1871
**(*n* = 43)**	*p*	0.4278	0.9428	0.0330 *	0.3243	0.2414
**GAL-1**	*r*	−0.0744	−0.0501	0.3603	−0.0479	0.3963
**(*n* = 28)**	*p*	0.7066	0.8000	0.0596	0.8089	0.0407 *
**IL-33**	*r*	−0.0099	−0.0549	0.4172	−0.0879	0.2478
**(*n* = 24)**	*p*	0.9634	0.7901	0.0340 *	0.6628	0.2127
**GAL-3**	*r*	0.0602	−0.3762	0.1957	−0.2102	0.1984
**(*n* = 30)**	*p*	0.7522	0.0405 *	0.2914	0.2564	0.3023

Spearman’s correlation coefficients (*r*) and corresponding *p*-values (*p*) are shown. Asterisks indicate nominal significance (unadjusted * *p* < 0.05). CPP = Chronic pelvic pain.

**Table 6 ijms-26-09581-t006:** Correlation between tumor markers and peripheral blood cells.

Blood Cells (10^3^/µL)		CA125 (*n* = 104)	CA19-9 (*n* = 93)	CEA (*n* = 82)	HE4 (*n* = 92)
**Leukocytes**	*r*	0.0462	−0.0948	0.2886	0.1947
	*p*	0.6415	0.3662	**0.0086** **	0.0629
**PMNs**	*r*	0.0884	−0.03774	0.1713	0.1390
	*p*	0.3719	0.7195	0.1240	0.1862
**Monocytes**	*r*	0.0369	0.0187	0.2626	0.1455
	*p*	0.7103	0.8590	**0.0172 ***	0.1665
**Blood cells** **(%)**					
**PMNs**	*r*	0.0992	0.0868	−0.0704	−0.0348
	*p*	0.3165	0.4079	0.5295	0.7423
**Monocytes**	*r*	−0.0388	0.1225	0.0375	0.0285
	*p*	0.6954	0.2420	0.7380	0.7872

Spearman’s correlation coefficients (*r*) and corresponding *p*-values (*p*) are shown. Asterisks indicate nominal significance (unadjusted * = *p* < 0.05, ** = *p* < 0.01). *p*-values surviving multiple testing correction using the Benjamini–Hochberg procedure (FDR < 0.05) are highlighted in bold.

**Table 7 ijms-26-09581-t007:** Correlation between tumor markers and inflammatory mediators.

Molecule		CA125	CA19-9	CEA	HE4
**TNF-α** **(*n* = 45–55)**	*r*	−0.0676	−0.0011	0.1518	0.1634
	*p*	0.6239	0.9934	0.2979	0.2834
**IL-1β** **(*n* = 28–32)**	*r*	0.2299	−0.0853	0.1889	−0.3259
	*p*	0.2055	0.6425	0.3089	0.0906
**IL-8** **(*n* = 65–74)**	*r*	0.3011	0.3172	0.0566	−0.1385
	*p*	**0.0091 ****	**0.0059 ****	0.6365	0.2713
**TGF-β** **(*n* = 30–37)**	*r*	0.0382	0.0679	0.1207	−0.1533
	*p*	0.8225	0.6898	0.4831	0.4186
**IL-10** **(*n* = 48–51)**	*r*	0.0387	0.0318	−0.0079	0.0990
	*p*	0.7876	0.8267	0.9564	0.5030
**sCD163**	*r*	−0.0662	−0.1514	−0.0536	−0.2180
**(*n* = 35–43)**	*p*	0.6734	0.3324	0.7395	0.2084
**GAL-1**	*r*	−0.3255	−0.1280	−0.1706	−0.2051
**(*n* = 23–26)**	*p*	0.1046	*0.5333*	0.4150	0.3479
**IL-33**	*r*	−0.5559	−0.1917	−0.3309	−0.3168
**(*n* = 27)**	*p*	**0.0026 ****	0.3380	0.0918	0.1148
**GAL-3**	*r*	0.2874	0.0336	−0.1517	−0.0870
**(*n* = 24–23)**	*p*	0.1460	0.8678	0.4594	0.6861

Correlation coefficients (*r*) and *p*-values were calculated using Pearson’s or Spearman’s tests, according to data distribution. Asterisks indicate nominal significance (unadjusted, ** = *p* < 0.01). *p*-values surviving multiple testing correction using the Benjamini–Hochberg procedure (FDR < 0.05) are highlighted in bold.

**Table 8 ijms-26-09581-t008:** Correlation between tumor marker and patient-reported pain.

Molecule		Dysmenorrhea	CPP	Dyschezia	Dysuria	Dyspareunia
**CA125** **(*n* = 97)**	*r*	0.1785	−0.0626	−0.2113	−0.1327	−0.1407
	*p*	0.0803	0.5404	0.0321 *	0.1792	0.1670
**CA19-9** **(*n* = 87)**	*r*	−0.0099	−0.2161	−0.0665	−0.0478	−0.2387
	*p*	0.9273	0.0408 *	0.5288	0.6491	**0.0251 ***
**CEA** **(*n* = 85)**	*r*	0.1070	−0.0401	0.1277	0.1399	0.0382
	*p*	0.3608	0.7276	0.2590	0.2129	0.7402
**HE4** **(*n* = 85)**	*r*	0.2211	−0.1511	−0.0151	0.0060	−0.0361
	*p*	0.0420 *	0.1575	0.8866	0.9546	0.7401

Spearman’s correlation coefficients (*r*) and corresponding *p*-values (*p*) are shown Asterisks indicate nominal significance (unadjusted * = *p* < 0.05). *p*-values surviving multiple testing correction using the Benjamini–Hochberg procedure (FDR < 0.05) are highlighted in bold. CPP = Chronic pelvic pain.

## Data Availability

The original contributions presented in this study are included in the article. Further inquiries can be directed to the corresponding author(s).

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
