# Peer review of "Systemic Immune and Tumor Marker Profiles in Ovarian and Deep Infiltrating Endometriosis: Associations with Disease Severity and Symptom Burden"

_ijms, 2025, doi:10.3390/ijms26199581_

Round 1
Reviewer 1 Report
Comments and Suggestions for Authors
In this manuscript, Ramírez-Pavez et al, discussed Systemic Immune and Tumor Marker Profiles in Ovarian and Deep Infiltrating Endometriosis: Associations with Disease Severity and Symptom Burden. This kind of study is valuable because it provides new insights into the immune and biomarker profiles of endometriosis. The work adds to our understanding of disease heterogeneity and may guide future development of non-invasive diagnostic tools. With some clarifications and refinements, the manuscript can make a strong contribution to the field.
- In the current study, most of the patients are stage IV or DIE. Authors need to discuss how this affects generalizability.
- Many patients and controls were on hormonal therapy. Authors need to show results stratified by hormone use.
- Too many correlations tested without adjustment. Please correct for multiple testing.
- Plasma was stored at –20 °C for years. Authors need to explain stability and handling of cytokines
- I suggest authors to maintain units (GAL-1, CA125) consistently throughout the manuscript
- IL-33 findings are mixed. Please expand discussion and explain the dual role.
- Tumor markers are non-specific. Please highlight diagnostic limitations clearly.
Author Response
- In the current study, most of the patients are stage IV or DIE. Authors need to discuss how this affects generalizability.
We thank the reviewer for this important observation. The predominance of advanced-stage and DIE cases in our cohort reflects the clinical profile of women undergoing laparoscopic surgery in our centers, where surgical intervention is usually indicated for severe disease, and also responds to the design of parallel projects in which peritoneal samples were collected. While this may limit the generalizability of our results to earlier stages of endometriosis, it provides a clinically homogeneous and well-characterized population, highly suitable for systemic immune and biomarker exploration. We have now expanded the Discussion to explicitly acknowledge this limitation and to clarify that studying advanced disease may also offer valuable insights into the robustness of candidate blood biomarkers under real-world clinical conditions, including hormonal treatment, and into their potential applicability for monitoring patients across different disease stages.
- Many patients and controls were on hormonal therapy. Authors need to show results stratified by hormone use.
We thank the reviewer for this valuable comment. Hormonal therapy was indeed common in our cohort (≈70% of patients and 40% of controls), reflecting real-world clinical practice. To address this concern, we performed exploratory stratified analyses by treatment status. These analyses showed that correlations between immune parameters and pain symptoms were more evident in untreated patients, while they were largely attenuated in those under hormonal therapy, consistent with the intended therapeutic effect of these treatments. However, the untreated subgroup was small, particularly for cytokine analyses, which markedly limited statistical power and led to loss of many associations after correction for multiple testing. For this reason, we chose not to present the full stratified data in the manuscript, but we now explicitly state in the Discussion and Conclusions that hormonal therapy is a potential confounder, and that larger treatment-naïve cohorts will be needed to validate these findings.
- Too many correlations tested without adjustment. Please correct for multiple testing.
We thank the reviewer for this important observation. Following the suggestion, we reanalyzed all correlation data applying the Benjamini–Hochberg procedure to control the false discovery rate (FDR). As a result, the significant correlation between rASRM score and neutrophil proportion (Figure 2C) remained robust after correction. In contrast, the inverse correlation between monocyte proportion and dysuria intensity (Table 2) did not reach significance once adjusted (original p = 0.025). We have now updated the figure and tables accordingly and specified in the Methods (section 4.5) that all correlation p-values were adjusted using the Benjamini–Hochberg procedure, with significance determined at FDR < 0.05.
- Plasma was stored at –20 °C for years. Authors need to explain stability and handling of cytokines
We thank the reviewer for pointing out this error. In the revised version, we will correct the description in the Materials and Methods section to clarify that plasma samples were stored at –80 °C, not –20 °C. Storage at –80 °C is consistent with current standards and ensures the stability of cytokines. We apologize for the oversight and have amended the text accordingly.
- I suggest authors to maintain units (GAL-1, CA125) consistently throughout the manuscript
We appreciate this observation. We have carefully revised the manuscript to ensure that units are reported consistently across text, figures, and tables. For example, GAL-1 is now uniformly expressed in ng/mL, and CA125 in U/mL.
- IL-33 findings are mixed. Please expand discussion and explain the dual role.
We thank the reviewer for noting this point. In our revised Results, only the association between IL-33 and CA125 remained significant after FDR correction, whereas correlations with rASRM and dyschezia did not withstand adjustment. We have expanded the Discussion to elaborate on the dual and context-dependent role of IL-33. Specifically, we highlight that while IL-33 can act as a pro-inflammatory alarmin promoting neutrophil recruitment, it also has regulatory properties that may dampen chronic inflammation. In our cohort, only the inverse association between IL-33 and CA125 remained significant after FDR correction, whereas correlations with rASRM severity and dyschezia did not withstand adjustment. This duality and context dependence may partly explain the divergent associations reported across studies.
7.Tumor markers are non-specific. Please highlight diagnostic limitations clearly.
We agree with the reviewer that tumor markers are nonspecific. We have revised the Discussion to emphasize their limited standalone diagnostic value and to clarify that, although subtle associations were observed with clinical variables, their utility lies in potential integration with immune profiles rather than as isolated biomarkers. We also now explicitly state that these findings are exploratory.
Reviewer 2 Report
Comments and Suggestions for Authors
The author studied the pathogenesis of endometriosis. The results show that the detection of peripheral blood may be of certain help for diagnosis. This is a supplement to radiological and ultrasound diagnosis and has certain clinical significance. There are the following points that the author needs to explain:
1.Regarding the structure of the article. "Materials and Methods" should be placed before "Results".
2.Pages 476-477. “It included 50 women undergoing laparoscopic tubal sterilization, who served as the control group.” Why were women in the control group selected as those who received laparoscopic tubal sterilization? What is the basis?
3.The abstract and results of the article indicate that the control group consists of 50 healthy women. If 50 women in the control group had received laparoscopic tubal sterilization, would this statement be appropriate? Or do you need special instructions? Also, what is the reason that 50 women received laparoscopic tubal sterilization? It needs to be clarified whether the surgery was performed due to a related disease.
Author Response
- Regarding the structure of the article. "Materials and Methods" should be placed before "Results".
We followed the journal’s submission guidelines, where the “Materials and Methods” section is placed after the “Results.” Unfortunately, the structure is not under our control.
- Pages 476-477. “It included 50 women undergoing laparoscopic tubal sterilization, who served as the control group.” Why were women in the control group selected as those who received laparoscopic tubal sterilization? What is the basis?
We selected women scheduled for elective laparoscopic tubal sterilization for contraception as the control group because they provide an optimal, healthy reference population: reproductive-age, fertile, and without gynecological disease. The laparoscopic setting also allowed exclusion of occult pelvic pathology, ensuring true absence of endometriosis. Controls were chosen from the same gynecology service and were comparable in age and clinical context to the endometriosis cohort, which reduces confounding. Importantly, no additional procedures were performed for research purposes; in this study we analyzed only peripheral blood obtained as part of routine perioperative care. Patients with endometriosis were included only when the diagnosis was confirmed at laparoscopy, both to ensure accurate case definition and because related peritoneal assessments were conducted in parallel projects (not reported here). Inclusion and exclusion criteria for controls mirrored those applied to cases—except for the absence of endometriosis—so that controls represent a valid reference for the endometriosis patients.
- The abstract and results of the article indicate that the control group consists of 50 healthy women. If 50 women in the control group had received laparoscopic tubal sterilization, would this statement be appropriate? Or do you need special instructions? Also, what is the reason that 50 women received laparoscopic tubal sterilization? It needs to be clarified whether the surgery was performed due to a related disease.
We appreciate the reviewer’s comment and the opportunity to clarify this point. The control group consisted of healthy women without gynecological disease or fertility problems. Peripheral blood samples were obtained in the preoperative phase, prior to surgery. Laparoscopic tubal sterilization was performed exclusively as a voluntary contraceptive procedure in the context of completed childbearing, and not for any pathological indication. For this reason, it is appropriate to describe them as “healthy controls” throughout the manuscript. Furthermore, we have revised Section 4.1 Patients in the Methods to explicitly clarify this aspect.
Reviewer 3 Report
Comments and Suggestions for Authors
The authors present a study examining systemic immune profiles and circulating tumor markers in patients with ovarian endometrioma and deep infiltrating endometriosis. The results reveal subtype-specific immune profiles: OE patients exhibit increased monocytes and relative neutrophil reduction, whereas DIE patients show elevated IL-8 and Galectin-1. Certain cytokines, notably IL-33 and TNF-α, correlate with disease severity and pain, while tumor markers such as CEA and CA125 reflect immune activation and lesion burden. This work is interesting, as it links immune alterations to disease severity, symptom burden, and tumor markers. While the study provides potentially valuable insights into subtype-specific immune patterns, some aspects of the methodology and data presentation could be clarified to strengthen the manuscript.
Major Concerns
1. A substantial proportion of patients (~70%) and a notable fraction of healthy controls (~40%) were receiving hormonal therapy at the time of sample collection. Hormonal treatment is known to significantly alter immune cell distributions, circulating cytokine levels, and tumor marker concentrations, which may confound the observed associations between systemic immune profiles, tumor markers, and disease features.
My suggestions:
-The authors should discuss this potential confounding effect more explicitly in the Discussion section and acknowledge its impact on interpretation of the results.
-If possible, subgroup analyses comparing patients with and without hormonal therapy, or statistical adjustment for hormone use as a covariate, would strengthen the validity of the conclusions.
-At minimum, the limitation regarding hormone exposure should be clearly stated in the Conclusions, emphasizing that some findings may reflect treatment effects rather than disease-specific immune alterations.
2. Most patients in this cohort were rASRM stage IV (67%) and had deep infiltrating endometriosis (DIE, 56.85%). This skewed sample may limit the generalizability of the findings to patients with milder disease and could overestimate the observed systemic immune and tumor marker alterations.
Minor concerns
1. Some experimental procedures are insufficiently detailed. For example, plasma processing is ambiguously described (“centrifugation for 3 minutes, 1300 rpm or 300 g”), and VAS pain assessment lacks standardization details. Clarification is needed to ensure reproducibility.
2. Some figures and text report slightly different sample sizes (e.g., Figure 1 control group n=47, methods section n=50). Why is it different?
3. The manuscript mentions hormonal therapy but does not detail the type, duration, or timing relative to sample collection, which could influence the observed immune and tumor marker profiles.
4. Other potential confounders, such as BMI differences, prior surgeries, or concomitant medications, are mentioned but not fully integrated into the analysis or discussion.
Author Response
Major Concerns
1. A substantial proportion of patients (~70%) and a notable fraction of healthy controls (~40%) were receiving hormonal therapy at the time of sample collection. Hormonal treatment is known to significantly alter immune cell distributions, circulating cytokine levels, and tumor marker concentrations, which may confound the observed associations between systemic immune profiles, tumor markers, and disease features.
My suggestions:
-The authors should discuss this potential confounding effect more explicitly in the Discussion section and acknowledge its impact on interpretation of the results.
-If possible, subgroup analyses comparing patients with and without hormonal therapy, or statistical adjustment for hormone use as a covariate, would strengthen the validity of the conclusions.
-At minimum, the limitation regarding hormone exposure should be clearly stated in the Conclusions, emphasizing that some findings may reflect treatment effects rather than disease-specific immune alterations.
We thank the reviewer for this important comment. Indeed, a substantial proportion of patients (~70%) and healthy controls (~40%) were under hormonal therapy at the time of sample collection. Hormonal therapy is expected to modulate immune cell activity, cytokine expression, and tumor marker levels. To address this, we have expanded the Discussion to explicitly acknowledge hormonal treatment as a potential confounding factor. We also conducted exploratory stratified analyses according to treatment status. As anticipated, associations between pain symptoms and circulating markers were more evident in untreated patients and attenuated in those receiving therapy. However, subgroup sizes were small, resulting in limited statistical power. For this reason, we present the stratified results only as exploratory, while keeping the main analyses on the full cohort, which we believe better reflects real-world clinical practice. We have also added a statement in the Conclusions explicitly noting that some findings may partly reflect treatment effects rather than disease-specific alterations, and that future studies in treatment-naïve cohorts will be required for validation.
Most patients in this cohort were rASRM stage IV (67%) and had deep infiltrating endometriosis (DIE, 56.85%). This skewed sample may limit the generalizability of the findings to patients with milder disease and could overestimate the observed systemic immune and tumor marker alterations.
We thank the reviewer for this observation. The predominance of advanced-stage (rASRM IV) and DIE cases in our cohort reflects the profile of patients undergoing laparoscopic surgery in our centers, where surgery is generally indicated for severe disease, and also stems from the design of parallel projects in which peritoneal samples and lesions were analyzed. While this distribution may indeed limit generalizability to milder disease stages, it provides a clinically homogeneous and well-characterized population, particularly suitable for systemic immune and biomarker exploration. We have revised the Discussion to explicitly acknowledge this limitation and to clarify that assessing advanced disease may still provide valuable insights into biomarker robustness and potential clinical applicability across different stages.
Minor concerns
1. Some experimental procedures are insufficiently detailed. For example, plasma processing is ambiguously described (“centrifugation for 3 minutes, 1300 rpm or 300 g”), and VAS pain assessment lacks standardization details. Clarification is needed to ensure reproducibility.
We thank the reviewer for noting this. We have revised the Materials and Methods section to clarify plasma processing and to detail VAS pain assessment, specifying the scale used and timing of evaluation. This should ensure reproducibility.
- Some figures and text report slightly different sample sizes (e.g., Figure 1 control group n=47, methods section n=50). Why is it different?
We apologize for this inconsistency. The control group initially included 50 women. For Figure 1, which reflects preoperative routine blood counts obtained from the clinical laboratory, complete data were available only for 47 controls. In addition, for cytokine and tumor marker analyses, not all plasma samples had sufficient volume to perform every determination, resulting in variable sample sizes across figures and tables. We have now corrected the text and figure legends for consistency, and—as indicated throughout the manuscript—the exact sample size (n) used for each measurement is specified in the corresponding figures and tables.
- The manuscript mentions hormonal therapy but does not detail the type, duration, or timing relative to sample collection, which could influence the observed immune and tumor marker profiles.
We thank the reviewer for this observation. In our cohort, hormonal therapy consisted mainly of oral contraceptives, progestins, or levonorgestrel-releasing intrauterine devices, prescribed for contraception in controls and for symptom management in patients. Importantly, these treatments were not suspended prior to surgery, when blood samples were collected, so all measurements reflect the ongoing influence of hormonal therapy. We believe that providing further details (e.g., duration) would not substantially change interpretation and could generate confusion, as it was not the focus of the present study. We have therefore clarified the type and indication of therapy in the Methods and noted in the Discussion that hormonal exposure represents a limitation for interpretation.
- Other potential confounders, such as BMI differences, prior surgeries, or concomitant medications, are mentioned but not fully integrated into the analysis or discussion.
We thank the reviewer for this observation. Potential confounders such as BMI, concomitant medication, or other comorbidities were controlled for by the inclusion and exclusion criteria of the study: all participants had a normal BMI, no relevant comorbidities, and were not taking concomitant medications. Therefore, these factors are unlikely to have influenced the systemic immune or tumor marker profiles observed. We have clarified this point in the Methods (section 4.1) to make it explicit.
Round 2
Reviewer 3 Report
Comments and Suggestions for Authors
Concerns have been addressed.